# Shaping the distribution of neural responses with interneurons in a recurrent circuit model

**David Lipshutz**
Center for Computational Neuroscience, Flatiron Institute
dlipshutz@flatironinstitute.org

**Eero P. Simoncelli**
Center for Computational Neuroscience, Flatiron Institute
Center for Neural Science, New York University
eero.simoncelli@nyu.edu

## Abstract

Efficient coding theory posits that sensory circuits transform natural signals into neural representations that maximize information transmission subject to resource constraints. Local interneurons are thought to play an important role in these transformations, dynamically shaping patterns of local circuit activity to facilitate and direct information flow. However, the relationship between these coordinated, nonlinear, circuit-level transformations and the properties of interneurons (e.g., connectivity, activation functions, response dynamics) remains unknown. Here, we propose a normative computational model that establishes such a relationship. Our model is derived from an optimal transport objective that conceptualizes the circuit's input-response function as transforming the inputs to achieve an efficient target response distribution. The circuit, which is comprised of primary neurons that are recurrently connected to a set of local interneurons, continuously optimizes this objective by dynamically adjusting both the synaptic connections between neurons as well as the interneuron activation functions. In an example application motivated by redundancy reduction, we construct a circuit that learns a dynamical nonlinear transformation that maps natural image data to a spherical Gaussian, significantly reducing statistical dependencies in neural responses. Overall, our results provide a framework in which the distribution of circuit responses is systematically and nonlinearly controlled by adjustment of interneuron connectivity and activation functions.

## 1 Introduction

The problem of transforming a signal into a representation with a given target distribution (or within a target set of distributions) is a classical problem whose origins can be traced back more than two centuries [1]. Many methods in statistics, signal processing and machine learning can be interpreted within this context. For example, data whitening is a common preprocessing step that linearly transforms a signal to have identity covariance [2]. Independent component analysis [ICA; 3] is a signal processing method that linearly transforms a signal so as to minimize higher-order statistical dependencies in addition to removing second-order dependencies. Nonlinear transformations of skewed or heavy-tailed data to approximately Gaussianize their distribution can facilitate statistical analyses [4, 5]. Machine learning methods for density estimation such as Gaussianization [6–10] and normalizing flows [11–13] nonlinearly transform high-dimensional signals with complex densities into more tractable representations with approximately Gaussian densities.

38th Conference on Neural Information Processing Systems (NeurIPS 2024).

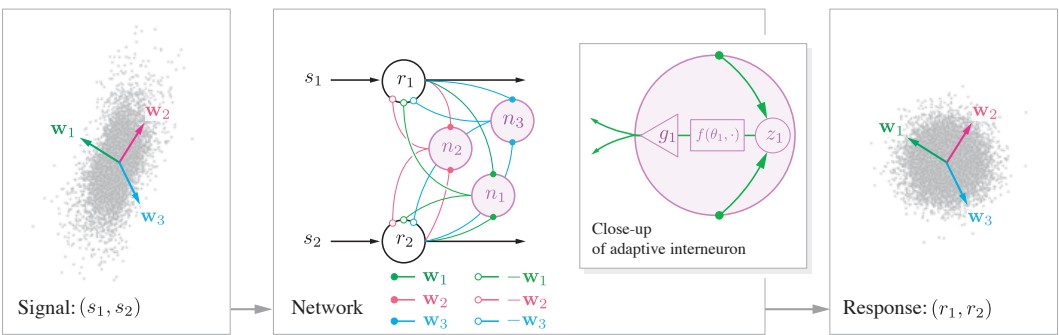

Figure 1: Schematic of a recurrent circuit with $N = 2$ primary neurons and $K = 3$ interneurons. **Left:** Scatter plot of a 2D input signals (gray points) $\mathbf{s} = (s_1, s_2)$ with $\mathbf{s} \sim p_{\mathbf{s}}$. **Center:** Primary neurons (black circles), with outputs $\mathbf{r} = (r_1, r_2)$, receive external feedforward inputs, $\mathbf{s}$, and recurrent feedback from an auxiliary population of interneurons (purple circles), $-\mathbf{W}\mathbf{n}$, where $\mathbf{n} = (n_1, n_2, n_3)$ are the interneuron outputs. Projection vectors $\{\mathbf{w}_1, \mathbf{w}_2, \mathbf{w}_3\}$ encode feedforward synaptic weights connecting primary neurons to interneurons $i = 1, 2, 3$, with symmetric feedback connections. **Inset:** The $i^{\text{th}}$ interneuron (here $i = 1$) receives weighted inputs $z_i := \mathbf{r} \cdot \mathbf{w}_i$, which is fed through the activation function $f(\theta_i, \cdot)$ and scaled by the gain $g_i$ to generate the output $n_i := g_i f(\theta_i, z_i)$. **Right:** Scatter plot of the 2D circuit responses $\mathbf{r} = (r_1, r_2)$ with $\mathbf{r} \sim p_{\text{target}}$.

This problem may also provide a framework for constructing and evaluating normative theories of sensory processing. Efficient coding theory posits that sensory systems maximize the information they transmit about sensory signals to downstream areas subject to resource constraints [14–17]. In one instantiation of this theory, the redundancy reduction hypothesis posits that sensory circuits transform natural signals into representations to minimize or eliminate statistical dependencies between coordinates, essentially producing factorized response distributions [14, 15, 18]. In a separate, but related, instantiation, sparse coding theory posits that population responses are optimized for sparsity [19–21], which is naturally interpreted as a constraint on the shape of the distribution of responses. In another line of theoretical work, sensory representations are posited to maximize the Fisher information of the inputs [22–24], which can be interpreted as a statement about the joint distribution of the inputs and responses. Importantly, each of these theories can be formulated as a transformation of the signal into a representation with target distribution that is optimal under information theoretic and metabolic constraints. However, it is not clear how neural circuits learn or implement these potentially nonlinear transformations.

Neural circuits are typically comprised of populations of primary (excitatory) neurons and local (inhibitory) interneurons. Extensive experimental measurements have led to the idea that local interneuron populations allow neural circuits to flexibly shape patterns of primary neuron responses so as to coordinate or regulate information flow [25–31]. Consequently, local interneurons are natural candidate mechanisms responsible for shaping circuit responses into efficient representations. However, the precise relationship between the physiological properties and connectivity of local interneurons and the coordinated response properties of populations of primary neurons remains unclear.

Several normative mechanistic models have been proposed to explain how neural circuits can *linearly* transform their inputs into a representation whose distribution lies within a target set (e.g., the set of distributions with identity covariance) [32–36]. These models are derived from optimization objectives for linear redundancy reduction, including (adaptive) decorrelation and ICA, and the circuit parameters (e.g., synaptic weights, response gains) are optimized to match the data distribution. In these cases, the optimization steps correspond to processes such as gain modulation and synaptic plasticity, thus demonstrating how adjustments of circuit parameters according to local signals can optimize a global, circuit-level objective for redundancy reduction of the neural responses. While linear transformations can remove second-order statistical dependencies, they cannot remove higher-order statistical dependencies that are prominent in sensory signals [7]. Furthermore, early sensory systems exhibit a host of prominent nonlinear response properties [37–39] that are not well-approximated by linear models.

Here, we seek a normative circuit model that *nonlinearly* transforms inputs to produce responses with a (spherical) target distribution. We develop an optimal transport objective for transforming the input signal into a neural representation with a given target distribution, and derive an algorithm (Alg. 1) that can be mapped onto a dynamical model of a neural circuit, Fig. 1. The circuit model is comprised of primary neurons that are recurrently connected to local interneurons and the circuit adapts its responses using a combination of Hebbian synaptic plasticity and interneuron adaptation. Complementary computational roles for Hebbian plasticity and interneuron adaptation emerge from this analysis: (i) synapses are updated according to a Hebbian learning rule to identify projections of the signal that are least aligned with the target distribution (essentially, a form of projection pursuit [40]); (ii) gains and nonlinear activation functions of interneurons are adjusted to transform the marginal circuit responses along directions defined by the synaptic weights. Together these operations transform the distribution of responses to approximate the target distribution.

As a primary test case motivated by redundancy reduction theory, we apply our algorithm to inputs derived from natural images (responses of local oriented filters, qualitatively similar to those found in the primary visual cortex) and the target distribution is the spherical Gaussian.[1] We find that the algorithm learns a nonlinear transformation that approximately Gaussianizes the responses and significantly reduces statistical dependencies between coordinates. Overall, our results demonstrate how local interneurons may adjust their connectivity and response properties to nonlinearly reshape the distribution of circuit responses, thus facilitating efficient transmission of information.

## 2 Circuit objective

Consider a neural circuit with $N \geq 2$ primary neurons that transforms input signals $\mathbf{s} \in \mathbb{R}^N$, which are distributed according to a density $p_\mathbf{s}$, into circuit responses $\mathbf{r} \in \mathbb{R}^N$ (Fig. 1). The inputs may represent a direct sensory input (e.g., the rate at which photons are absorbed by a cone) or the weighted sum of multiple inputs (e.g., the postsynaptic current). The responses $\mathbf{r}$ represent the firing rate (or the logarithm of the firing rate) of the neuron. For simplicity, we assume that the circuit responses are a deterministic function of the input signals; that is, $\mathbf{r} = T(\mathbf{s})$ for a function $T : \mathbb{R}^N \mapsto \mathbb{R}^N$.

### 2.1 Optimal transport objective

We assume that the objective of the circuit is to transform its inputs $\mathbf{s}$ so that the circuit responses $\mathbf{r} = T(\mathbf{s})$ follow a (spherical) target distribution $p_{\text{target}}$ while minimizing the $L^2$-distance between responses and input signals. Mathematically, this corresponds to an optimal transport problem [42]

$$\min_T \mathbb{E}\left[\|T(\mathbf{s}) - \mathbf{s}\|^2 + \lambda\|T(\mathbf{s})\|^2\right] \quad \text{such that} \quad T(\mathbf{s}) \sim p_{\text{target}}, \tag{1}$$

where the minimization is over a suitable class of functions $T$, $\lambda \in \mathbb{R}$ is a regularizing term and, unless otherwise noted, expectations are over the input distribution $p_\mathbf{s}$. Note that the choice of $\lambda$ does not affect the optimal solution as $\mathbb{E}[\|T(\mathbf{s})\|^2]$ is fixed provided $T(\mathbf{s}) \sim p_{\text{target}}$; however, it will affect the optimization algorithm. Assuming $p_\mathbf{s}$ is sufficiently regular, the minimum in eq. (1) is the squared Wasserstein-2 distance between $p_\mathbf{s}$ and $p_{\text{target}}$ and the input-response transformation $T$ is the so-called optimal transport plan that maps the input distribution $p_\mathbf{s}$ to the target distribution $p_{\text{target}}$.

Our goal is to derive an online algorithm for optimizing the objective in eq. (1) that can be implemented in a neural circuit model. We accomplish this by defining a distance between the response distribution $p_\mathbf{r}$ and the target distribution $p_{\text{target}}$ that can be estimated in a neural circuit, and then solving the optimization problem by incorporating this measure as a constraint, using the method of Lagrange multipliers.

### 2.2 Measuring the discrepancy between the response distribution and the target distribution

Common candidates for measuring the discrepancy between two distributions include Kullback-Leibler (KL) divergence or integral probability metrics [43]. The former typically require numerous

---

[1]Biologically, we can interpret this as a Gaussian distribution of voltages on cell bodies, which are transformed through an exponential activation function to a log-normal distribution of firing rates that are positive-valued, heavy-tailed, and similar to those observed in cortex [41].

samples to estimate the density $p_\mathbf{r}$, whereas neural circuits must operate in the online setting without access to the full history of their responses. Therefore, we use an integral probability metric that quantifies the difference between the random variables when evaluated using constraint functions that can be rapidly estimated online.

We restrict our solution to use constraint functions that compare the *marginal* response distributions to the marginals of $p_\text{target}$, denoted $p_\text{marginal}$, which are all equal under our assumption that $p_\text{target}$ is spherical. Such constraint functions are well matched to signals that are linear mixtures of independent sources, as in generative models for ICA. Even when the signal statistics are not generated according to a linear mixture model, the Cramér and Wold theorem [44] suggests that transforming sufficiently many marginals of the response distribution may effectively transform the multivariate response distribution (though the number of marginals required may be quite large). Our motivation for measuring marginal response distributions is due, in part, by our goal of modeling local interneurons, whose inputs are naturally modeled as weighted sums of primary neuron responses (i.e., their input distributions are marginals of the primary responses).

To compare the marginal response distributions, we first select a finite set of directions defined by $K \geq 1$ unit vectors $\mathbf{w}_1, \ldots, \mathbf{w}_K \in \mathbb{R}^N$, which can be either randomly sampled, chosen based on prior knowledge of the signal statistics, or learned from data using projection pursuit [40]. We then choose a class of scalar functions $\{h(\theta, \cdot)\}$ parameterized by $\theta$, which defines a semi-metric between the marginal of $\mathbf{r}$ in the direction $\mathbf{w}$ and $p_\text{marginal}$ to be $\max_\theta |\mathbb{E}[\phi(\theta, \mathbf{r} \cdot \mathbf{w})]|$, where

$$\phi(\theta, z) := h(\theta, z) - \mathbb{E}_{z \sim p_\text{marginal}}[h(\theta, z)].$$

For example, when $\{h(\theta, \cdot)\}$ parameterizes all Lipschitz-1 (indicator) functions, this induces the Wasserstein-1 (total variation) distance between the marginal response distribution and $p_\text{marginal}$. Given directions $\mathbf{w}_1, \ldots, \mathbf{w}_K$ and constraint functions $\{h(\theta, \cdot)\}$, we express the distance between the response density $p_\mathbf{r}$ and the standard Gaussian distribution as the sum of the marginal distances:

$$d_{\mathbf{W}, \phi}(p_\mathbf{r}) := \sum_{i=1}^{K} \max_{\theta_i} |\mathbb{E}_{\mathbf{r} \sim p_\mathbf{r}} [\phi(\theta_i, \mathbf{r} \cdot \mathbf{w}_i)]|,$$

where $\mathbf{W} := [\mathbf{w}_1, \ldots, \mathbf{w}_K]$ is the $N \times K$ matrix of concatenated unit vectors.[2]

How do we choose the constraint functions $\{h(\theta, \cdot)\}$? In general, the choice should be well-suited to the input distribution $p_\mathbf{s}$ and the target distribution $p_\mathbf{r}$. For example, consider the simple case when the inputs follow a centered Gaussian distribution with unknown covariance structure, and the target distribution is the spherical Gaussian distribution $\mathcal{N}(\mathbf{0}, \mathbf{I})$. The Wasserstein-2 distance between a marginal distribution of the inputs and the standard normal distribution $\mathcal{N}(0, 1)$ can be expressed in terms of the difference between the second moments, so a quadratic function $h(z) = \frac{1}{2}z^2$ suffices. In section 4.2, we consider a parametric class motivated by natural signal statistics.

## 2.3 Optimization using Lagrange multipliers

We replace the condition $\mathbf{r} \sim p_\text{target}$ in eq. (1) with the condition $\max_\mathbf{W} d_{\mathbf{W}, \phi}(p_\mathbf{r}) = 0$, which we enforce using Lagrange multipliers. This results in the minimax optimization problem

$$\max_\mathbf{W} \max_{\boldsymbol{\theta}} \max_\mathbf{g} \mathbb{E} \left[ \min_\mathbf{r} \mathcal{L}(\mathbf{W}, \boldsymbol{\theta}, \mathbf{g}, \mathbf{s}, \mathbf{r}) \right], \tag{2}$$

where $\mathcal{L}$ is defined by

$$\mathcal{L}(\mathbf{W}, \boldsymbol{\theta}, \mathbf{g}, \mathbf{s}, \mathbf{r}) := \|\mathbf{r} - \mathbf{s}\|^2 + \lambda\|\mathbf{r}\|^2 + \sum_{i=1}^{K} g_i \phi(\theta_i, \mathbf{r} \cdot \mathbf{w}_i), \tag{3}$$

and $\boldsymbol{\theta} := (\theta_1, \ldots, \theta_K)$ is the set of concatenated parameters, $\mathbf{g} := (g_1, \ldots, g_K)$ is a $K$-dimensional vector of Lagrange multipliers, and the circuit transform is defined by $T(\mathbf{s}) = \arg\min_\mathbf{r} \mathcal{L}(\mathbf{W}, \boldsymbol{\theta}, \mathbf{g}, \mathbf{s}, \mathbf{r})$. The maximization over $\mathbf{g}$ and $\boldsymbol{\theta}$ effectively minimizes the distances between the marginal distributions of the responses $\mathbf{r}$ along the directions $\mathbf{w}_1, \ldots, \mathbf{w}_K$ and $p_\text{marginal}$, whereas the maximization over the matrix $\mathbf{W}$ learns directions along which the marginals of $\mathbf{s}$ are least aligned with $p_\text{marginal}$, essentially performing projection pursuit [40].

---

[2]This distance is closely related to (max-)sliced Wasserstein metrics [45, 46], which quantify the distance between two distributions in terms of the Wasserstein distances between their marginal distributions. Specifically, when $K = 1$ and $h(\theta, \cdot)$ parameterizes the set of all Lipschitz-1 functions, then the sliced and max-sliced Wasserstein-1 distances between $p_\mathbf{r}$ and $p_\text{target}$ are $\mathbb{E}_{\mathbf{w} \sim \text{Unif}(S^{N-1})}[d_{\mathbf{w}, \phi}(p_\mathbf{r})]$ and $\max_\mathbf{w} d_{\mathbf{w}, \phi}(p_\mathbf{r})$.

# 3 Algorithm and circuit implementation

We now derive an online gradient-based algorithm for optimizing the objective in eq. (2), then map the algorithm onto a recurrent neural circuit. Spiking activity operates on a much faster timescale than neural or synaptic adaptation mechanisms, so we assume that the neural activities equilibrate before the neural activations and synapses are updated.

## 3.1 Fast recurrent neural dynamics

At each iteration, the circuit receives a stimulus $\mathbf{s}$. The (discretized) recurrent neural response dynamics (Fig. 1) correspond to gradient-descent minimization of $\mathcal{L}$ with respect to $\mathbf{r}$:

$$\mathbf{r} \leftarrow \mathbf{r} + \eta_r \left( \mathbf{s} - \mu\mathbf{r} - \sum_{i=1}^{K} n_i\mathbf{w}_i \right), \qquad n_i = g_i f(\theta_i, z_i). \tag{4}$$

where $\eta_r > 0$ is a small constant, $\mu := 1 + \lambda$ represents a leak term, $z_i := \mathbf{r} \cdot \mathbf{w}_i$ is the weighted input to the $i^{\text{th}}$ interneuron, $f(\theta_i, \cdot) := \partial\phi(\theta_i, \cdot)/\partial z$ is the activation function, $g_i$ is a multiplicative gain that scales the output, and $n_i$ denotes the output. Notably, the interneuron activation response function $g_i f(\theta_i, \cdot)$ is parameterized by $(g_i, \theta_i)$, which can vary across interneurons, so the interneuron responses are heterogeneous. For each $i$, synaptic weights $\mathbf{w}_i$ connect the primary neurons to the $i^{\text{th}}$ interneuron and symmetric weights $-\mathbf{w}_i$ connect the $i^{\text{th}}$ interneuron to the primary neurons. From eq. (4), we see that the neural responses are driven by the signal $\mathbf{s}$, a leak term $-\mathbf{r}$, and recurrent weighted feedback from the interneurons $-\mathbf{W}\mathbf{n}$, where $\mathbf{n} := (n_1, \ldots, n_K)$.

Since the neural activities equilibrate before other updates are performed, the responses $\mathbf{r}$ are a fixed point of $\mathcal{L}(\mathbf{W}, \boldsymbol{\theta}, \mathbf{g}, \mathbf{s}, \cdot)$. In general, we do not have a closed-form expression for $\mathbf{r}$; however, if $g_i \geq 0$ and $\phi(\theta_i, \cdot)$ are convex, then $\mathcal{L}(\mathbf{W}, \boldsymbol{\theta}, \mathbf{g}, \mathbf{s}, \cdot)$ is convex and we can express the transform as

$$T(\mathbf{s}) = \arg\min_{\mathbf{r}} \mathcal{L}(\mathbf{W}, \boldsymbol{\theta}, \mathbf{g}, \mathbf{s}, \mathbf{r}). \tag{5}$$

In Appx. A we show that the transform $T(\cdot)$ is invertible whenever $g_i \geq 0$ and $\phi(\theta_i, \cdot)$ are convex and thus it defines a precise relationship between the input distribution $p_\mathbf{s}$ and response distribution $p_\mathbf{r}$.

## 3.2 Gain modulation, activation function adaptation and Hebbian plasticity

After the neural activities reach equilibrium, we maximize $\mathcal{L}$ by taking concurrent gradient-ascent steps with respect to $g_i$, $\theta_i$ and $\mathbf{w}_i$:

$$\Delta g_i = \eta_g \phi(\theta_i, z_i), \qquad \Delta\theta_i = \eta_\theta \nabla_\theta \phi(\theta_i, z_i), \qquad \Delta\mathbf{w}_i = \eta_w n_i \mathbf{r},$$

where $\eta_g, \eta_\theta, \eta_w \geq 0$ are the respective learning rates, which control the relative speeds of gain modulation, neural adaptation and synaptic plasticity, respectively. For example, at the extremes, we can *fix* the gains, activation functions or synaptic weights by setting $\eta_g = 0$, $\eta_\theta = 0$ or $\eta_w = 0$, respectively. Notably, while synaptic plasticity [47] and gain modulation [48] are well-studied circuit mechanisms that support learning and adaptation, adjustments of nonlinear neural activation functions are not as well established. Nevertheless, there is some emerging evidence that neurons also adapt their activation functions in response to changes in their input statistics [49, 50].

The circuit operates online and the updates are local in the sense that the updates to the gain and activation function of the $i^{\text{th}}$ interneuron depend only on variables $\theta_i$ and $z_i$. The updates to the synapses $\mathbf{w}_i$ (and $-\mathbf{w}_i$) are proportional (inversely proportional) to the product of the pre- and postsynaptic activities, so they are both local and *Hebbian (anti-Hebbian)* [51]. Finally, to ensure that the vectors $\mathbf{w}_1, \ldots, \mathbf{w}_K$ have unit norm, we normalize the weights after each update: $\mathbf{w}_i \leftarrow \mathbf{w}_i/\|\mathbf{w}_i\|$. This can be viewed as form of homeostatic plasticity such as synaptic scaling [52]. While the feedforward weights $\mathbf{w}_i$ and feedback weights $-\mathbf{w}_i$ are constrained to be symmetric, these can be decoupled due to the symmetry of the Hebbian learning rule. Both theoretical and empirical evidence of this is shown for a related adaptive whitening circuit in [36, appendix E.2].

## 3.3 Online algorithm

Combining the neural dynamics, the interneuron adaptation and synaptic plasticity steps yields our online algorithm (Alg. 1), which we write in vector-matrix notation by defining the normalization function $P(\mathbf{W}) := [\mathbf{w}_1/\|\mathbf{w}_1\|, \ldots, \mathbf{w}_K/\|\mathbf{w}_K\|]$.

**Algorithm 1:** Approximate optimal transport with Hebbian plasticity and interneuron adaptation

---

1: **input:** $\mathbf{s}_1, \mathbf{s}_2, \ldots$
2: **initialize:** $\mathbf{W}, \boldsymbol{\theta}, \mathbf{g}, \mu, \eta_r, \eta_g, \eta_\theta, \eta_w$
3: **for** $t = 1, 2, \ldots$ **do**
4:     $\mathbf{r}_t \leftarrow \mathbf{s}_t$
5:     **while** not converged **do**
6:         $\mathbf{z}_t \leftarrow \mathbf{W}^\top \mathbf{r}_t$ ;                        // interneuron inputs
7:         $\mathbf{n}_t \leftarrow \mathbf{g} \circ f(\boldsymbol{\theta}, \mathbf{z}_t)$ ;                        // interneuron outputs
8:         $\mathbf{r}_t \leftarrow \mathbf{r}_t + \eta_r (\mathbf{s}_t - \mu \mathbf{r}_t - \mathbf{W} \mathbf{n}_t)$ ;                        // neural responses
9:     **end while**
10:     $\mathbf{g} \leftarrow \mathbf{g} + \eta_g \phi(\boldsymbol{\theta}, \mathbf{z}_t)$ ;                        // gain update
11:     $\boldsymbol{\theta} \leftarrow \boldsymbol{\theta} + \eta_\theta \nabla_\theta \phi(\boldsymbol{\theta}, \mathbf{z}_t)$ ;                        // activation update
12:     $\mathbf{W} \leftarrow P(\mathbf{W} + \eta_w \mathbf{r}_t \mathbf{n}_t^\top)$ ;                        // Hebbian + homeostatic plasticity
13: **end for**

---

### 3.4 Relation to existing algorithms

Algorithm 1 is naturally viewed as a nonlinear extension of a number of existing algorithms for linear data whitening that have neural circuit implementations [34, 36, 53, 54]. In particular, when the constraint function is quadratic, $h(z) = \frac{1}{2}z^2$, and the target distribution is the spherical Gaussian, $\mathcal{N}(\mathbf{0}, \mathbf{I})$, then the activation function is the identity, $f(z) = z$, and the optimization in eq. (2) enforces that the second moments of the responses match the second moments of $\mathcal{N}(\mathbf{0}, \mathbf{I})$, which corresponds to data whitening. If the gains are fixed ($\eta_g = 0$) and $K \geq N$ (i.e., synaptic adaptation only), then Alg. 1 corresponds to the adaptive whitening algorithm presented in [34, 53]. Alternatively, if the synaptic weights are fixed ($\eta_w = 0$) and $K \geq N(N+1)/2$ (i.e., interneuron gain adaptation only), then Alg. 1 corresponds to the adaptive whitening algorithm presented in [54]. Finally, if the gains adapt on a fast timescale and the synapses update on a slow timescale (i.e., $\eta_g \gg \eta_w > 0$), Alg. 1 corresponds to the multi-timescale adaptive whitening algorithm presented in [36].

When the target distribution is the spherical Gaussian and $\mathbf{W}$ is constrained to be an orthogonal matrix, Alg. 1 is related to existing iterative algorithms for Gaussianization that alternate between (a) orthogonal transformations and (b) marginal Gaussianization of the coordinates [6, 8]. In the case that the column vectors of $\mathbf{W}$ are orthogonal, Gaussianization along one marginal does not affect the responses along other marginals, allowing these operations to be performed independently of one another. In general, we allow the column vectors of $\mathbf{W}$ to be non-orthogonal and potentially overcomplete so that marginal Gaussianization along one basis vector affects the other marginal distributions. Consequently, the algorithm is more complicated to analyze mathematically (e.g., obtaining convergence guarantees), but it is more biologically realistic since neural systems are unlikely to have orthonormal synaptic weight vectors.

## 4 Gaussianization of natural image statistics

We apply our algorithm to the problem of efficient nonlinear encoding of natural signals, specifically oriented filter responses to visual images.[3] Redundancy reduction theories posit that early sensory systems transform natural signal into neural representations with reduced statistical redundancies [14, 15, 17]. In support of this hypothesis, early sensory representations exhibit far less spatial and temporal correlation than natural signals [55, 56] and methods such as linear ICA have been used to derive optimal representations of natural signals that are approximately matched to early sensory neural properties [18, 19, 57].

Linear whitening and ICA transforms can eliminate simple forms of statistical dependency, but their responses exhibit higher-order statistical dependencies when applied to natural signals [58], suggesting that sensory systems can more efficiently represent natural signals by implementing nonlinear transforms. Consistent with this, nonlinear phenomenological models of neural responses (e.g., divisive normalization [37, 39]) effectively reduce these higher-order statistical dependencies [38, 59]. However, the circuit mechanisms that support these *nonlinear* transformations are unknown.

---

[3]Example code for our experiments can be found at `https://github.com/dlipshutz/shaping`.

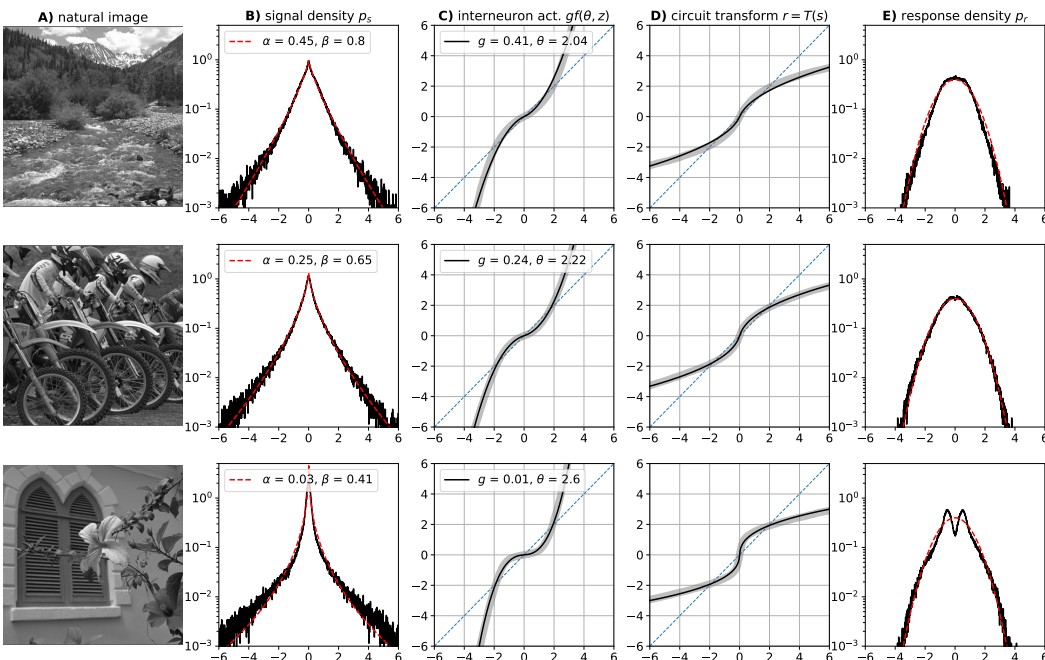

Figure 2: Marginal Gaussianization of local filter responses. **A)** Three example natural images from the Kodak dataset. **B)** Histograms of local filter responses (black lines) and fitted generalized Gaussian density (red dashed lines) with scale $\alpha$ and shape exponent $\beta$. **C)** Learned interneuron activations $gf(\theta, z)$, with $f(\theta, z)$ defined as in eq. (6) and learned $g$ and $\theta$, and **D)** corresponding stimulus-response transforms $r = T(s)$. The optimal activations and transforms are shown as thick gray curves. **E)** Histograms of the circuit responses (black lines) and the Gaussian density (red dashed lines).

We consider the particular target of Gaussian responses, which may be interpreted as firing rates, or as their logs (i.e.., membrane potentials are Gaussian, and firing rates are exponentiated). From the perspective of coding efficiency and computation, Gaussian representations are appealing for a variety of reasons. First, among distributions with given covariance structure, Gaussian distributions have maximum entropy. Therefore, if metabolic demands are a function of the (co)variance of the response distribution, then the Gaussian distribution maximizes information transmission under metabolic constraints. Second, compression based on information theoretic objectives often reduces to linear projection when the data distribution is Gaussian [60–62], so Gaussianization facilitates downstream computation. In addition, the efficiency of the representation is preserved under orthogonal transformation [7]. Finally, experiments have shown that single neurons in the fly early visual system adaptively Gaussianize their univariate responses [63], and neural populations in early sensory systems decorrelate their multivariate responses [56, 64–67]. Therefore, neural circuit models that nonlinearly transform signals to jointly Gaussianize their responses may offer normative, parsimonious explanations of nonlinear transformations in early sensory systems.

## 4.1 Description of the input signal

We computed the responses of a local oriented filter [68], roughly matched to typical receptive field selectivity of neurons in primate visual cortex [69], applied to natural images from the Kodak dataset [70]. These local filter responses are notorious for their sparse heavy-tailed statistical properties that can be well approximated by *generalized Gaussian* distributions of the form $p_s(s) \propto \exp(-|s/\alpha|^\beta)$, where $\alpha$ is referred to as the *scale* parameter and $\beta$ is referred to as the *shape* parameter [71]. Fig. 2AB shows example images and histograms of the local filter responses along with fitted generalized Gaussian distributions whose scale and shape parameters $(\alpha, \beta)$ vary across images. While linear methods such as variance normalization are sufficient for rescaling the distribution, adaptive nonlinear transformations are required to reshape these heavy-tailed distributions.

Next, we generated 2-dimensional signals from pairs of the local filter responses for images at fixed horizontal spatial offsets ranging between $d = 2$ and $d = 64$. Contour plots (using kernel density

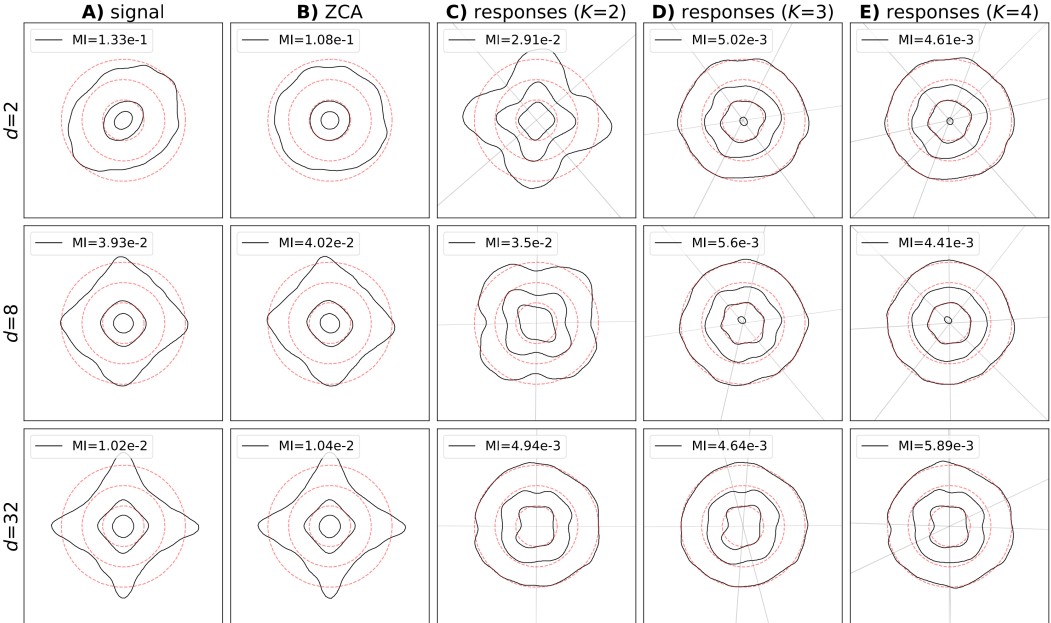

Figure 3: Joint Gaussianization of pairs of filter responses, spatially displaced by $d = 2, 8, 32$ pixels. Evenly spaced contours for the spherical Gaussian distribution are depicted as dashed red circles. Iso-probability contours for **(A)** the local filter responses, **(B)** ZCA whitened local filter responses, and **(CDE)** learned circuit responses (with $K = 2, 3, 4$ interneurons) are depicted (in the respective column) as black curves along with the estimated mutual information between coordinates. The learned column vectors of $\mathbf{W}$ are indicated by the faint gray lines.

estimation) of the local filter response pairs and symmetric (or ZCA) whitened local filter response pairs for a natural image (specifically, the top-left image from Fig. 2) are shown for select $d$ in Fig. 3AB. To quantify the statistical dependencies between coefficients, we estimated the mutual information between the pairs of coefficients after discretizing them into bins of width 0.5. In Fig. 4, we plot the estimated mutual information (using bin size 0.5) between the local filter response pairs (blue line) and ZCA whitened local filter response pairs (orange line) for spatial offsets between $d = 2$ and $d = 64$. Note that aside from $d = 2$, the linear ZCA whitening transform does not significantly reduce the mutual information between coordinates. (Similar results have been found when applying linear ICA transforms; see, e.g., [72, Figure 6].) Fig. 4 also suggests that ZCA whitening can even slightly increase the mutual information between coordinates—see also, rows $d = 8$ and $d = 32$ of Fig. 3AB—though these effects are quite small and may be a consequence of the discretization step when estimating mutual information.

## 4.2 Choice of activation functions

How do we choose the family of activation functions $\{f(\theta, \cdot)\}$? One approach is to choose a kernel that can approximate a general class of functions. An alternative approach, adopted here, which is motivated by the efficient coding hypothesis [14, 15, 17], is to choose a family of activation functions that is well matched to the marginal statistics of natural signals. In Fig. 2C, we plot examples of optimal activations for transforming local filter responses from different images (thick gray curves).

Since the marginals of the local filter responses are well-approximated by generalized Gaussian distributions [71], a sensible approach is to identify a family of interneuron activation functions that are optimal for transforming generalized Gaussian distributions with varying $(\alpha, \beta)$ into the standard Gaussian distribution. When $\mu = 0$, this implies (see Appx. B) that for each choice of scale $\alpha$ and shape $\beta$, there is a gain $g$ and parameter $\theta$ such that

$$gf(\theta, \cdot) = F_{\alpha,\beta}^{-1} \circ \Phi(\cdot),$$

where $\Phi(\cdot)$ is the cdf of $\mathcal{N}(0, 1)$. However, if we define $f(\theta, z)$ in terms of the above display, then we do not have a closed-form solution for $\phi(\theta, z)$ or $\nabla_\theta \phi(\theta, z)$, which are both required to implement

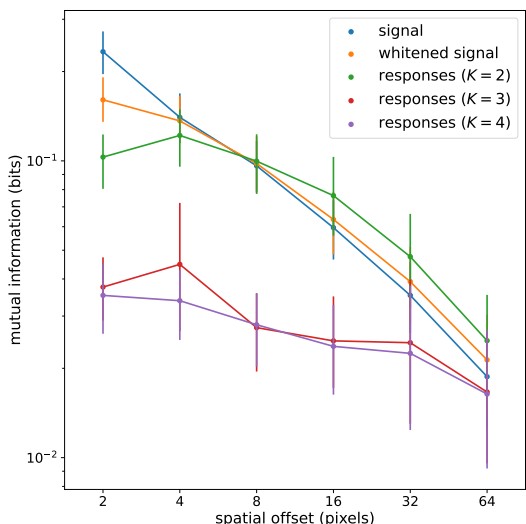

Figure 4: Estimated mutual information, with 95% confidence intervals (estimated across 23 images), for original signals (pairs of filter responses), the ZCA whitened signal, and learned circuit responses (with $K = 2, 3, 4$ interneurons).

Alg. 1. Instead, we found that $F_{\alpha,\beta}^{-1} \circ \Phi(\cdot)$ can be well approximated by the simple algebraic form

$$f(\theta, z) = a(\theta)z + b(\theta)\operatorname{sign}(z)|z|^{\theta}, \tag{6}$$

where $a(\theta)$ and $b(\theta)$ are specified nonnegative functions of $\theta > 1$. Intuitively, the linear component shapes the marginal density locally around zero, while the higher-order monomial shapes the tails of the marginal distribution. In Appx. B, we show that the monomial activation $f(\theta, z) = \operatorname{sign}(z)|z|^{\theta}$ is optimal for Gaussianizing scalar signals whose marginal tail densities satisfy $p_s(s) \propto |s|^{q-1}\exp(-|s|^{2q})$, where $q = 1/\theta$. This closely resembles the tail densities of generalized Gaussian densities (when $\alpha = 1$), suggesting monomial activations are effective for shaping the tails.

### 4.3 Marginal density of local filter responses

We first apply Alg. 1 in the scalar setting $N = K = 1$ to demonstrate that our choice of activation function in eq. (6) is indeed well matched to the shape of heavy-tailed marginals of local filter responses. For each image, we ran Alg. 1 on the local filter responses with $\mu = 0$, learning rates $(\eta_g, \eta_\theta) = (10^{-5}, 10^{-5})$ and batch size 10 for $10^5$ iterations. Fig. 2CD shows the learned interneuron activation functions $gf(\theta, \cdot)$ and the learned transforms $T(\cdot)$. Fig. 2E shows histograms of the circuit responses. Compared to the local filter responses, the circuit responses are visually much closer to Gaussian. We found that the circuit performs worse when the distribution $p_s$ is more 'peaked' around zero (i.e., when $s$ is sparser and the fitted shape parameter $\beta$ is smaller), as evidenced by the mismatch between the response distribution $p_r$ and the Gaussian distribution $\mathcal{N}(0, 1)$ near zero in the bottom row of Fig. 2E (see Appx. C for more examples). However, even in this case, the interneuron activation and circuit transform are close to optimal (Fig. 2CD, bottom row).

### 4.4 Joint density of pairs of local filter responses

Next, we apply Alg. 1 in the multivariate setting $N = 2$, to pairs of spatially offset filter responses. For each image and offset, we ran Alg. 1 with $K = 2, 3, 4$ interneurons, $\mu = 0$, learning rates $(\eta_g, \eta_\theta, \eta_w) = (10^{-4}, 10^{-6}, 10^{-4})$, and batch size 10 for $10^6$ iterations. Contour plots of the learned circuit responses for one image are shown in Fig. 3CDE; see Appx. C for more examples. The mutual information between circuit responses is shown in Fig. 4. We see that $K = 3$ interneurons significantly reduces the mutual information between circuit responses for spatial offsets less than $d = 32$. The reduction is much greater than obtained using $K = 2$ interneurons and about the same as obtained using $K = 4$ interneurons. For spatial offsets greater than $d = 32$, the raw filter responses already have low mutual information and the circuit does not not offer any significant reduction.

# 5 Discussion

We derived a novel online algorithm for transforming a signal to approximate a target distribution, using a recurrent neural circuit with Hebbian synaptic plasticity, gain modulation, and adaptation of interneuron activation functions. The model draws inspiration from the extensive neuroscience literature on efficient coding [15], Hebbian synaptic plasticity [47], interneuron function [29] and gain modulation [48], proposing complementary roles for different physiological processes: Hebbian synaptic plasticity learns stimulus axes that are least matched to the target distribution, and interneurons adapt their gains and activation functions to transform the marginal responses along these axes. The form of the input-output function for the local interneurons—linear-nonlinear with gain modulation—closely resembles phenomenological models of neurons [73], and the parameters $\mathbf{W}$, $\boldsymbol{\theta}$, $\mathbf{g}$ can potentially be fit to neural recordings and compared with the optimal parameters that can be derived from the signal statistics $p_\mathbf{s}$. Furthermore, our model predicts a relationship between the interneuron activation function and the circuit transform; see Fig. 2CD for an example of an expansive interneuron activation function that corresponds to a compressive circuit transformation.

There are, however, some aspects of our circuit that are not biologically realistic. For example, our model focuses on the role of local interneurons in reshaping the response distribution and, for simplicity, assumes that the primary neurons have linear activation functions. A more realistic model would also include nonlinearity and adaptation in the primary neurons. Moreover, our model only includes synaptic connections between primary neurons and interneurons, which may be consistent with some sensory circuits (e.g., olfactory bulb), but cannot account for excitatory-excitatory connections or inhibitory-inhibitory connections in cortical circuits. Finally, the synaptic weights are not sign-constrained, and violate Dale's law. This can be addressed by modifying the objective in eq. 2 so that the optimization is over non-negative weight matrices $\mathbf{W} \geq 0$, which will result in a projected gradient step in Alg. 1; however, the circuit responses will generally be less aligned with the target distribution.

A limitation of our simulations is that we only test our method on two-dimensional inputs, demonstrating that three interneurons are sufficient to dramatically reduce the redundancy in the circuit responses. However, natural signals are generally high-dimensional and it is not clear how the number of interneurons required to effectively reduce redundancy will scale with the dimension of the signal. There is some basis for optimism. For example, visual inputs are highly structured—e.g., statistical dependencies between inputs rapidly decay with the distance between the inputs—so local interneurons only need to connect to neurons with overlapping or adjacent receptive fields, limiting the number of interneurons that are required as the dimensionality of the input signal grows [54].

There are a number of existing computational models that also explain how neural circuits can implement nonlinear transformations to efficiently encode their inputs. For example, several neural circuit models implement forms of divisive normalization [74–76], a transformation that is optimal for efficient encoding of natural signals [38, 59]. In addition, there is a body of work on normative spiking models derived from objectives that maximize the information encoded per spike [77–79], which can account for neural adaptation mechanisms such as gain control. Our work differs from these, by proposing a novel framing of sensory circuit computation in terms of transformations of probability distributions, which can be viewed as a population level version of the seminal work by Laughlin [16]. We then demonstrate in a normative circuit model how interneurons can play a critical role in optimizing this objective by measuring the marginal distribution of circuit responses and adjusting their feedback accordingly.

Finally, our results may also be relevant beyond the biological setting. Gaussianization and normalizing flows are active areas of research [10, 13, 80]. We offer a novel continuous-learning solution inspired by neuroscience that learns using a combination of weight updates and activation function updates (related to trainable activation functions [81]). In low-dimensional settings, when the constraint functions are matched to the signal statistics, we show that Gaussianization can be achieved using relatively few parameters. It is of primary interest to understand how the methods introduced here scale to high-dimensional signals, where the curse of dimensionality presents significant challenges.

## Acknowledgments

We thank Colin Bredenberg and members of the Center for Computational Neuroscience at the Flatiron Institute for helpful feedback on an earlier draft of this work.

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

# A Analysis of circuit transform

In this section, we analyze the circuit transform $T(\cdot)$ in the case that $\mu \geq 0$ (i.e., $\lambda \geq -1$), $g_i \geq 0$ and $\phi(\theta_i, \cdot)$ are convex so that $\mathcal{L}(\mathbf{W}, \boldsymbol{\theta}, \mathbf{g}, \mathbf{s}, \cdot)$ is convex and

$$T(\mathbf{s}) = \arg\min_{\mathbf{r}} \mathcal{L}(\mathbf{W}, \boldsymbol{\theta}, \mathbf{g}, \mathbf{s}, \mathbf{r}).$$

When $\mu > 0$, we can solve for the equilibrium responses using the fixed point iteration:

$$\mu \mathbf{r}^{(0)} = \mathbf{s}, \qquad\qquad \mu \mathbf{r}^{(n+1)} = \mathbf{s} - \mathbf{W}(\mathbf{g} \circ f(\boldsymbol{\theta}, \mathbf{W}^\top \mathbf{r}^{(n)})),$$

where $f(\boldsymbol{\theta}, \mathbf{z}) := (f(\theta_1, z_1), \ldots, f(\theta_K, z_K))$ denotes the vector obtained by applying the function $f$ elementwise to the pairs $(\theta_i, z_i)$ and '$\circ$' denotes the elementwise (Hadamard) product.

## A.1 Invertibility of the transform

If the constraint functions are twice continuously differentiable, then $T(\mathbf{s})$ is continuously differentiable and we can relate the response density $p_\mathbf{r}$ to the signal density $p_\mathbf{s}$ as follows:

$$p_\mathbf{s}(\mathbf{s}) = |\det(J_T(\mathbf{s}))| \, p_\mathbf{r}(T(\mathbf{s})),$$

where $J_T$ denotes the Jacobian of $T$ with respect to $\mathbf{s}$. In general, we don't have a closed form solution for $T$; however, setting the update in eq. (4) to zero, we see that the inverse transform $T^{-1}$ (when it is well-defined) satisfies

$$T^{-1}(\mathbf{r}) = \mathbf{s} = \mu \mathbf{r} + \sum_{i=1}^{K} g_i f(\theta_i, \mathbf{r} \cdot \mathbf{w}_i) \mathbf{w}_i, \tag{7}$$

where we have used the fact that the partial derivative of $\phi(\theta, z)$ with respect to $z$ is equal to the partial derivative of $h(\theta, z)$ with respect to $z$. It follows that its Jacobian $J_{T^{-1}}$ satisfies

$$J_{T^{-1}}(\mathbf{r}) = \mu \mathbf{I} + \mathbf{W} \mathrm{diag}\left(g_1 a_1(\mathbf{r}), \ldots, g_K a_K(\mathbf{r})\right) \mathbf{W}^\top,$$

where $a_i(\mathbf{r}) := \partial^2 \phi(\theta_i, \mathbf{r} \cdot \mathbf{w}_i)/\partial z^2$. Provided that either (a) $\mu > 0$ or (b) $g_i > 0$ and $\phi(\theta_i, \cdot)$ is strictly convex for all $i$ and the column vectors of $\mathbf{W}$ span $\mathbb{R}^N$, then the Jacobian of $T^{-1}$ is positive definite everywhere. This implies that the Jacobian of $T$ is positive definite everywhere, and thus $T$ is invertible everywhere.

## A.2 Relation to function class

Equation (7) establishes a relationship between the set of parameterizable (inverse) transforms and the function class $\{h(\theta, \cdot)\}$. To better understand this relationship, consider the scalar setting $N = K = 1$ in which case the optimal transformation is given by $T = F_{\mathrm{marginal}}^{-1} \circ F_s$, where $F_{\mathrm{target}}$ and $F_s$ are the cumulative distribution functions (cdfs) of $p_{\mathrm{marginal}}$ and $p_s$, respectively. When $T = F_{\mathrm{marginal}}^{-1} \circ F_s$, it follows from eq. (7) that $h(\theta, z)$ satisfies

$$g \frac{\partial h(\theta, z)}{\partial z} = g f(\theta, z) = F_s^{-1} \circ F_{\mathrm{marginal}}(z) - \mu z. \tag{8}$$

The multidimensional setting is more complicated; however, in the case that the signal is an orthogonal mixture of $N$ independent sources, we can derive a precise relationship between the signal density and $(\mathbf{W}, \boldsymbol{\theta}, \mathbf{g})$ Consider the case that the signal is of the form $\mathbf{s} = \mathbf{A}\mathbf{u}$, where $\mathbf{A}$ is an $N \times N$ orthogonal mixing matrix and $\mathbf{u} = (u_1, \ldots, u_N)$ has statistically independent coordinates. Suppose $\mathbf{W} = \mathbf{A}^\top$. After left multiplying eq. (7) on both sides by $\mathbf{W}^\top$, we get

$$\mathbf{u} = \mu \mathbf{W}^\top \mathbf{r} + \mathbf{g} \circ \mathbf{f}(\boldsymbol{\theta}, \mathbf{W}^\top \mathbf{r}).$$

If $\mathbf{r}$ is Gaussian, then so is $\mathbf{z} = \mathbf{W}^\top \mathbf{r}$ and

$$u_i = \mu z_i + g_i f(\theta_i, z_i), \qquad\qquad i = 1 \ldots, N.$$

Therefore, an optimal solution to eq. (2) is when $\{\theta_i\}$ and $\{g_i\}$ satisfy

$$g_i f_i(\theta_i, z_i) = F_{u_i}^{-1} \circ F_{\mathrm{marginal}}(z_i) - \mu z_i, \qquad i = 1, \ldots, N, \tag{9}$$

where $F_{u_i}$ is the cumulative distribution function for $u_i$.

# B   Activation functions

In this section, we discuss the activation functions used for the experiments carried out in Sec. 4 of the main text.

## B.1   Choice of activation functions

How should we choose the family of activation functions $\{f(\theta, \cdot)\}$? As stated in the main text, one approach is the choose a family that is well matched to the marginal statistics of the inputs. In this case, the input marginals are well approximated by generalized Gaussian distributions of the form

$$p_s(s) = \frac{\beta}{2\alpha\Gamma(1/\beta)} \exp(-|s/\alpha|^\beta)$$

with varying scale $\alpha$ and shape $\beta$. Here $\Gamma$ is the gamma function. Therefore, from eq. (8), we see that an optimal family of activation functions $\{f(\theta, \cdot)\}$ is such that for each choice of scale $\alpha$ and shape $\beta$, there is a gain $g$ and parameter $\theta$ such that

$$gf(\theta, z) + \mu z = F_{\alpha,\beta}^{-1} \circ \Phi(z),$$

where $F_{\alpha,\beta}$ and $\Phi$ denote the cdfs of the generalized Gaussian distribution and the standard Gaussian distribution. In general, defining the family of activation functions directly in terms of the above display leads to challenges implementing Alg. 1 since $\phi(\theta, z)$ and $\nabla_\theta \phi(\theta, z)$ are not readily computable.

We instead sought a simple algebraic expression that approximates $F_{\alpha,\beta}^{-1} \circ \Phi(z)$. First, suppose the activation function takes the form $f(\theta, z) = \text{sign}(z)|z|^\theta$. When $\mu = 0$, this activation function is optimal for a signal whose cdf $F_{\text{monomial}}$ satisfies

$$g\text{sign}(z)|z|^\theta = F_{\text{monomial}}^{-1} \circ \Phi(z).$$

Rearranging and differentiating with respect to $z$, we find that the pdf of the signal $p_{\text{monomial}}$ satsifies

$$p_{\text{monomial}}(g\text{sign}(z)|z|^\theta)g\theta|z|^{\theta-1} = \frac{1}{\sqrt{2\pi}} \exp\left(-\frac{1}{2}z^2\right).$$

Substituting in with $s = g\text{sign}(z)|z|^\theta$, we find the following expression for the pdf

$$p_{\text{monomial}}(s) = \frac{q}{g\sqrt{2\pi}}|s/g|^{q-1} \exp\left(-\frac{1}{2}|s/g|^{2q}\right),$$

where $q = 1/\theta$. This closely resembles aspects of the pdf for the *tails* of the generalized Gaussian distribution. To reshape the distribution local when $s \approx 0$, we included a linear term, resulting in the activation function

$$f(\theta, z) = a(\theta)z + b(\theta)\text{sign}(z)|z|^\theta.$$

Here $a(\theta)$ and $b(\theta)$ are nonnegative functions given by

$$a(\theta) = \exp((2\theta - 3.85)^{1.95})$$
$$b(\theta) = \exp(\theta^{2.32} - 5.9).$$

The forms of $a(\theta)$ and $b(\theta)$ were chosen to approximately minimize $\min_\theta \max_z |F_s(gf(\theta, z)) - \Phi(z)|$ when $F_s$ is the cdf of a generalized Gaussian distribution with shape parameter $\beta$ between 0.2 and 1.

## B.2   Activation function updates

Here we derive the updates to the activation functions. Recall from Alg. 1 that the $\theta$ update is given by $\theta \leftarrow \theta + \eta_\theta \nabla_\theta \phi(\theta, z)$, where

$$\phi(\theta, z) := h(\theta, z) - \mathbb{E}_{z \sim \mathcal{N}(0,1)}[h(\theta, z)] = \frac{a(\theta)}{2}(z^2 - 1) + \frac{b(\theta)}{\theta + 1}(|z|^{\theta+1} - C(\theta + 1)),$$

and $C(p)$ is the absolute $p$-moment of a scalar Gaussian random variable:

$$C(p) := \mathbb{E}_{z \sim \mathcal{N}(0,1)}[|z|^p] = \sqrt{\frac{2^p}{\pi}} \Gamma\left(\frac{p+1}{2}\right).$$

Note that when $\theta = 3$, the constraint function normalizes the kurtosis of the marginal and the resulting update bears similarity to ones used in neural models of ICA [82]. Differentiating with respect to $\theta$, we obtain the updates

$$\frac{\partial \phi(\theta, z)}{\partial \theta} = \frac{a'(\theta)}{2}(z^2 - 1) + \frac{(\theta+1)b'(\theta) - b(\theta)}{(\theta+1)^2}(|z|^{\theta+1} - C(\theta+1))$$

$$+ \frac{b(\theta)}{\theta+1}\left(|z|^{\theta+1}\log|z| - C'(\theta+1)\right),$$

where

$$C'(p) = \frac{1}{2}\sqrt{\frac{2^p}{\pi}}\Gamma\left(\frac{p+1}{2}\right)\left(\log 2 + \psi^{(0)}\left(\frac{p+1}{2}\right)\right), \tag{10}$$

$\psi^{(0)}(p)$ is the polygamma function and we have used the fact that the derivative of the gamma function is $\Gamma'(p) = \Gamma(p)\psi^{(0)}(p)$.

## C Experiments

### C.1 Experimental set up

We ran our experiments on a cluster comprised of 40-core Intel Skylake nodes with 768GB of RAM. Each experiment shown in Fig. 2 (i.e., Gaussianization of the coefficients from 1 image) took less than 5 minutes to run. Each experiment shown in Fig. 3 took around 5 hours to run (we did not optimize the choice of hyper-parameters).

### C.2 Additional experimental results

In Fig. 5, we provide additional examples of histograms of local filter responses and optimized responses for various images from the Kodak dataset [70] (see Fig. 2 of the main text for a detailed description). We note that when the shape parameter $\beta$ of the fitted generalized Gaussian distribution is small, the distribution of responses has a characteristic dip near zero. This is likely due to the fact that when $\beta$ is small, there is more probability mass concentrated near zero and so small discrepancies between the learned activation function $gf(\theta, z)$ and the optimal activation function $F_{\alpha,\beta}^{-1} \circ \Phi(z)$ when $z \approx 0$ can lead to large discrepancies between the response distribution and $\mathcal{N}(0, 1)$ near zero. This could potentially be resolved by choosing a parameterization of $gf(\theta, z)$ that better approximates the optimal activation function.

In Fig. 6, we provide additional examples of contour plots of local filter responses and optimized responses for three additional images from the Kodak dataset (corresponds to top three rows of Fig. 5, see Fig. 3 of the main text for a detailed description). In some of these examples we see that unlike the standard Gaussian distribution $\mathcal{N}(\mathbf{0}, \mathbf{I})$ the learned response distribution is clearly non-monotone with respect to $\|\mathbf{r}\|$. This non-monotonicity is likely inherited from non-optimality of the interneuron activation functions that leads to the discrepancies in the scalar case (see Fig. 5).

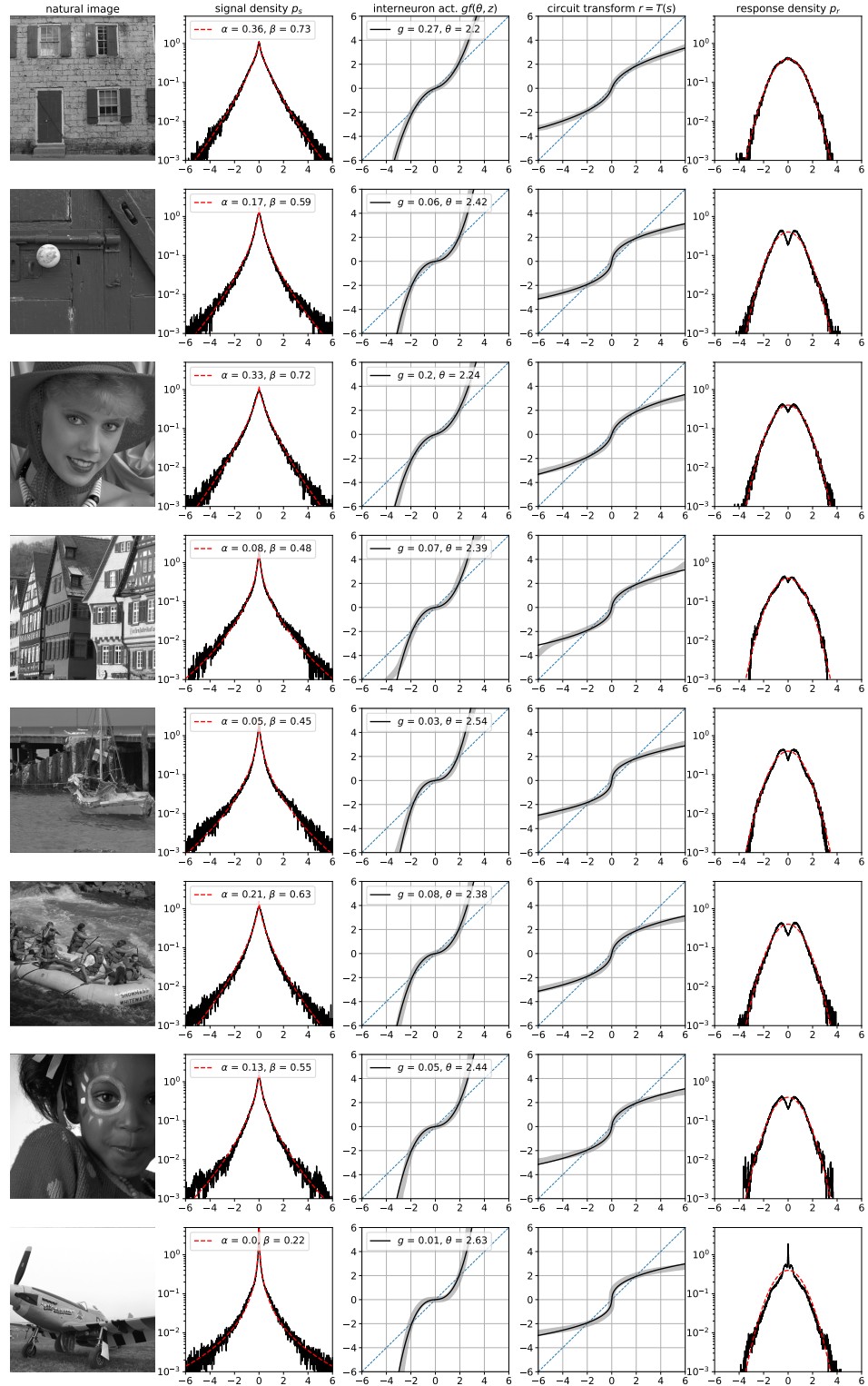

Figure 5: Gaussianization of local filter responses.

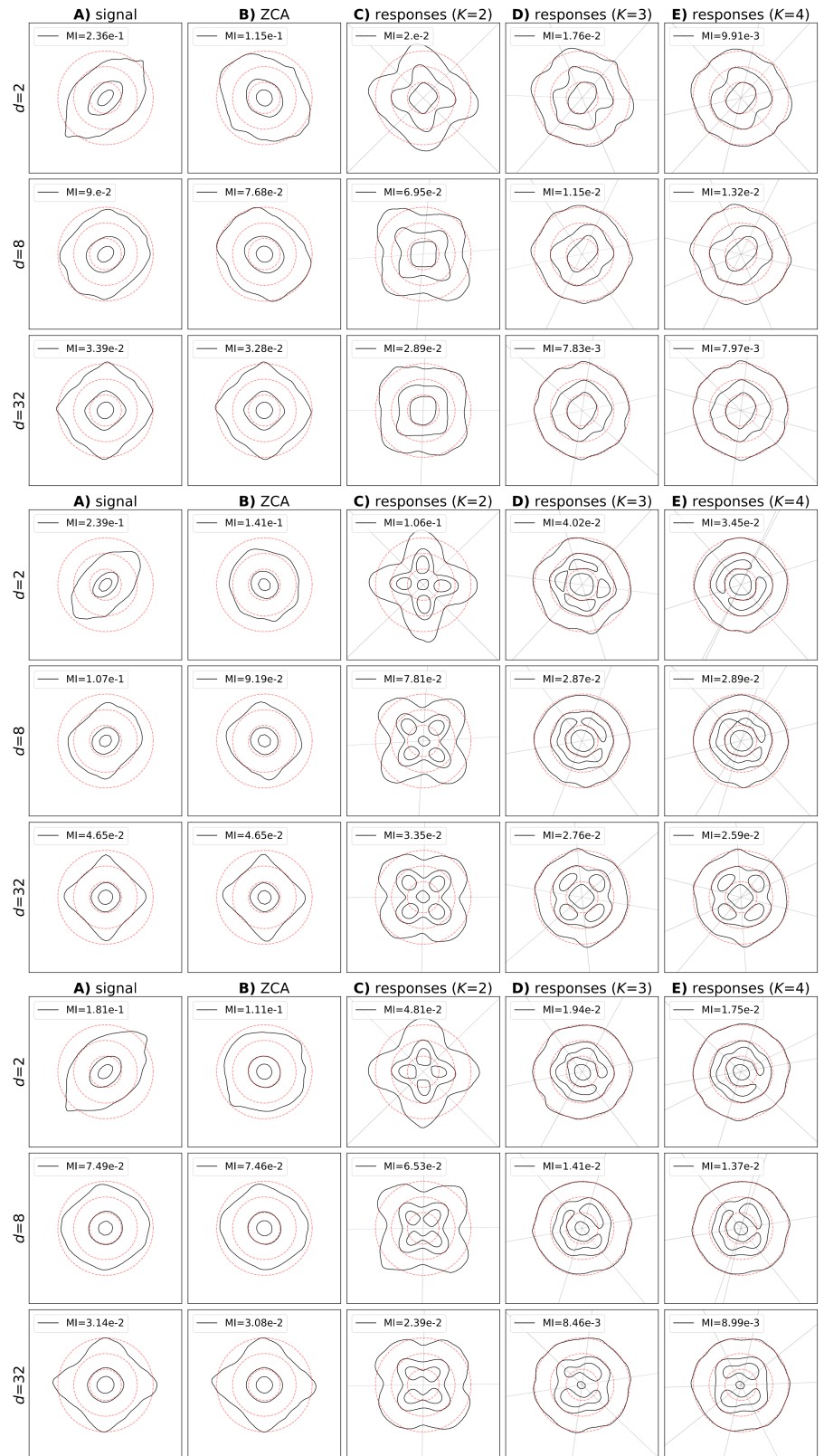

Figure 6: Gaussianization of pairs of local filter responses.

