# OpenReview forum: "Shaping the distribution of neural responses with interneurons in a recurrent circuit model"
_NeurIPS.cc/2024/Conference — NeurIPS 2024 poster_

### Official Review · Reviewer_DTUH · 2024-06-14

**Soundness:** 3
**Presentation:** 4
**Contribution:** 4
**Rating:** 8
**Confidence:** 4

**Summary:**

This paper proposes a normative recurrent circuit to solve an optimal transport problem, focusing on the problem of Gaussianization of natural image statistics.

**Strengths:**

This paper was a pleasure to read. The writing is clear, the formulation of the problem elegant, and the results represent a clear advance relative to past works on whitening circuits.

**Weaknesses:**

I have only a few small concerns regarding the biological realism of the model, but these are mostly already acknowledged by the authors. One point that I believe warrants further discussion is the plausibility of the activation functions (6). These are somewhat reminiscent of two-sided versions of the rectified-power law activations used in cortical models, no? I also have a handful of miscellaneous questions (see below), but on the whole these do not dampen my enthusiasm.

**Questions:**

- Along with the work of Van Hateren, the authors might consider citing Laughlin, "A simple coding procedure enhances a neuron's information capacity," and Juusola and Hardie, "Light Adaptation in Drosophila Photoreceptors." I leave it to their discretion whether or not to do so.

- In that vein, it might be nice to provide further evidence for adaptation of activation functions in biology. If additional references occur to me, I will update my review.

- In Line 271, should "activations" be plural? I think this is a typo.

- The legibility of Figure 3A could be improved; as it stands the text is a bit too small and the lines a bit too faint.

- It could be nice to provide further support for the approximate form of the activation functions. Can you numerically evaluate the expression below Line 630?

**Limitations:**

As noted above, the authors clearly discuss the limitations of their work.

---

> ### Author Rebuttal · Authors · 2024-08-05
>
> Thank you for your review, we are pleased that you enjoyed reading our work!
>
> ### Weaknesses
>
> We agree the plausibility of the activation functions warrants further discussion. The interneuron activation functions indeed resemble two-sided rectified-power law activations. A potentially more plausible variant would have interneurons responding with a  *rectified* power law, although implementing the same computation would then require twice as many interneurons.
>
> Alternatively, our model provides a precise relation between the input distribution, target distribution and interneuron activation function. Therefore, fixing two of these determines the third. We fixed the input distribution and target distribution and derived the optimal interneuron activation function. One could instead fix the input distribution and interneuron activation function (e.g., based on experimental measurements) and estimate the target distribution.
>
> ### Questions
>
> - Thank you for the suggestion. We will include references to the works by Laughlin and Juusola & Hardie in our revision.
> - We will also add a reference to Sharpee et al. "Adaptive filtering enhances information transmission in visual cortex", which showed that neurons in cat primary visual cortex adapt to higher-order statistics beyond mean and covariance.
> - Thanks for catching this typo.
> - We will improve the legibility of Figure 3A.
> - Yes, numerical evaluations of the expression below Line 630 are shown as thick gray curves in Figure 2C. To emphasize this, we will add text below Line 630 pointing the reader to the figure.

---

> > ### Comment · Reviewer_DTUH · 2024-08-07
> >
> > Thanks for your thoughtful response to my comments and those of the other reviewers. I maintain my positive assessment.
> >
> > A comment: I agree with Reviewer Voaj that it is important to acknowledge past work on normalization and predictive coding with spiking networks. However, I am familiar with these works, and in my opinion the mathematical framing of the present manuscript is certainly sufficiently novel and interesting to warrant publication. I'm not too concerned with violations of Dale's law; this is already an interesting first step.

---

### Official Review · Reviewer_txmi · 2024-06-25

**Soundness:** 3
**Presentation:** 2
**Contribution:** 2
**Rating:** 5
**Confidence:** 2

**Summary:**

This paper investigates a crucial question in neuroscience: how do neural circuits convert inputs into a target distribution, specifically focusing on how local interneurons transform natural image statistics into a spherical Gaussian distribution. The authors approach this problem through the lens of optimal transport. By using an integral probability metric, the task of transforming the input distribution into a spherical Gaussian distribution is framed as a minimax optimization problem. This constrained minimax optimization translates to recurrent neural dynamics (representing the minimization part) combined with gain modulation, activation function adaptation, and plasticity (representing the maximization part), with plasticity being local and Hebbian. Notably, the study's principled adjustment of activation functions is a unique contribution. The authors present experimental results on natural images using wavelet transformations.

**Strengths:**

Note: As I am not well-versed in wavelet transformations, which are extensively utilized in the experimental section, my review focuses primarily on the theoretical aspects prior to the experiments (Section 4).

**Originality**: This paper tackles a significant question in neuroscience: how neural circuits transform input signals into a target distribution. The authors frame this problem as an optimal transport problem. Utilizing the integral probability metric, they demonstrate that this problem is equivalent to a constrained minimax optimization, which can be interpreted through neural circuit dynamics. The paper's theoretical contributions are both solid and original.

**Quality**: The work is theoretically robust, with well-justified methods and experiments that effectively validate the theoretical model. The use of spherical Gaussian distributions, which might initially appear to be a limitation, is convincingly justified in Section 4.

**Clarity**: The paper is clearly written overall.

**Significance**: This paper introduces a normative model for normalizing inputs, offering potentially valuable insights for the theoretical neuroscience community.

**Weaknesses:**

- The authors should define $p_{target}$, $p_{marginal}$, and $p_r$ explicitly instead of using descriptive terms for better clarity.
- The proposed model necessitates different $g$, $\theta$, and $w$ for different stimuli. While neurons can potentially adjust their gain and activation function (i.e., $g$, $\theta$) relatively quickly, the need for weight updates to handle varying inputs significantly limits the model's applicability and biological plausibility.
- As noted in the limitations section, the weights are not sign-constrained. Therefore the distinguishing between excitatory and inhibitory neurons in this model makes no sense.

**Questions:**

The constraint $g_i > 0$ is not included in Algorithm 1. Would imposing these constraints ($w > 0$, $g > 0$) negatively impact the experimental results?

**Limitations:**

The authors have adequately addressed the limitation.

---

> ### Author Rebuttal · Authors · 2024-08-06
>
> Thank you for your comments. We will revise our paper in accordance with your suggestions.
>
> ### Weaknesses
> - We will explicitly define these distributions in our revision.
> - This is a great point. Interestingly, the learned weights $W$ are approximately shared across images, whereas the optimal $g$ and $\theta$ vary between images, suggesting that there are structural properties that are shared across images. This is consistent with rapid adjustments of $g$ and $\theta$ and slow adjustments of $W$. We did focus on this point because Duong et al. [NeurIPS 2023] have shown similar results. Specifically, they showed how a multi-timescale (linear) circuit with fast gain adjustments and slow synaptic updates can effectively whiten circuit responses to changing contexts. Here, our focus is on the nonlinear aspects of the circuit so we did not emphasize this point, but we will note it in our revision.
> - We can enforce $W>0$ at the cost of a slight degradation in the performance of the algorithm; see our general author response above and the attached PDF.
>
> ### Questions
> Indeed, we did not include the constraint $g_i>0$ in Algorithm 1. However, the optimal solutions all have $g_i>0$ and including the constraint would not have changed the experimental results. As mentioned above, enforcing $W>0$ results in a slight degradation in performance.

---

> > ### Comment · Reviewer_txmi · 2024-08-09
> >
> > I've read the other reviewers' comments and appreciate the authors' thoughtful responses. I'll be keeping my score unchanged.

---

### Official Review · Reviewer_Voaj · 2024-07-11

**Soundness:** 3
**Presentation:** 2
**Contribution:** 2
**Rating:** 5
**Confidence:** 3

**Summary:**

Authors propose an online algorithm that solves the problem of optimal transport. In particular, assuming a spherical distribution of stimuli, the goal of the algorithm is to generate neural responses such that their distribution best approximates the distribution of stimuli. Authors find that a neural network with excitatory and inhibitory neurons can solve such optimization problem, by learning a non-linear transformation that reduces dependency between neural responses. This is done through Hebbian learning on synapses and by adjusting activation functions of single neurons.

**Strengths:**

The paper presents is a novel combination of well known techniques and provides an alternative to existing approaches. The work seems technically sound and authors describe some interesting relation to similar approaches. Proposed algorithm is a nonlinear extension of existing algorithms for data whitening using a neural network.

**Weaknesses:**

The paper has the ambition of finding a neural implementation of generalized whitening such that it could be also implemented by biological neural circuits. However, a major concern is that inhibitory neurons do not obey Dale's law, and the model is therefore not biologically plausible. In general, it is not clear what the paper brings to understanding of biological or artificial neural networks.
Since the model violates Dale's law, the discussion about the function of interneurons seems out of scope.

Moreover, there are significant parts of the paper that are unclear. A number of details about the methods are given, but the main mechanism that supports learning remains unclear. Authors mention that the circuit "learns directions that are least matched to the target distribution", but it remained unclear to me how is this achieved and why this helps to Gaussianize the distribution of output firing rates of principal neurons.

Authors motivate their model by claiming that divisive normalization has not yet been solved with a neural network. This seems false (see for example Chalk, Masset, Deneve & Gutkin,  PLOS CB 2017). Gain control is also captured by available models. Efficient spiking models and normative models where changes in gain control can be captured by modulating the metabolic cost on spiking in the model's objective (see Guiterrez and Deneve, eLife 2019; Koren and Deneve, PLOS CB 2017). In general, the paper does not sufficiently take into account previous work on efficient coding with spikes that is closely related.

There are typos on several places, for example: lines 75, 77, 100, 234

**Questions:**

Authors motivate their modelling by claiming that it is currently unclear how neural networks could implement nonlinear transformations. Which non-linear transformations are not captured by the listed models?
The paper by Alemi et al. AAAI 2018, proposes efficient spiking models that implement non-linear transformation of stimuli at the population level. In Koren & Panzeri, NeurIPS 2022, neural populations perform linear transformation of stimuli on the level of neural populations, but on the level of single neurons these transformations are strongly non-linear. What do authors mean by non-linear transformations? Non-linear on what level? This should be clarified.

Authors say that approximating target distribution is useful because it facilitates efficient transmission of signals. It is however not entirely clear why making the distribution of responses Gaussian-like facilitates efficient transmission. Is this improvement in efficient transmission attributed to redundancy reduction?

Is inhibition required to solve this optimization problem, e.g., could the same operation be achieved through plasticity of E neurons?

**Limitations:**

Authors addressed the limitations adequately.

---

> ### Author Rebuttal · Authors · 2024-08-06
>
> Thank you for your careful reading of our paper. We take your concerns quite seriously and have revised our paper accordingly. Please find our responses to your listed weakness and your questions below.
>
> ### Weaknesses
> - We tested a modified version of our algorithm in which we enforce Dale's law. The modified version performs well, although there is a noticeable degradation in performance when compared with the original algorithm. For more details, please see our general author response and the attached PDF.
> - We appreciate your feedback that parts of the paper are unclear. We'll do our best to improve the revised description.
> - Thank you for pointing out these omissions. We take this quite seriously and plan to revise our paper to make it clear that there are a number of existing nonlinear neural circuits for efficiently encoding their inputs (see our general author response).
> - Thank you for pointing out these typos.
>
> ### Questions
> - To be precise, we mean that the circuit transform is nonlinear. Specifically, the function $T:{\bf s}\mapsto{\bf r}$, which maps the vector of circuit inputs to the vector of circuit responses, is nonlinear. We will clarify this in our revision.
> - Yes, the primary reason is attributed to redundancy reduction. There are other factors that make Gaussian distributions appealing, but this is the main reason.
> - It is likely that plasticity of excitatory neurons can contribute to reshaping of neural responses; however, redundancy reduction likely requires local inhibitory interneurons. For example, we are unaware how plasticity of EE connections can reduce correlations between two primary neuron responses that receive highly correlated inputs.

---

> > ### Comment · Reviewer_Voaj · 2024-08-08
> >
> > Thank you for your reply to my questions. In light of improvements made in the rebuttal I am increasing my score to 5.
> >
> > I appreciate that you implemented and tested the model that obeys Dale's law - this is an important proof of concept in spite of  reduced performance of the Dalian network. Also, I find that your work can be better appreciated in light of clarifications and improvements described in the general rebuttal. I suggest that these improvements and clarifications are carefully incorporated into the revised paper, in particular, the points about the batch training and symmetry of synaptic weights seem important.
> >
> > I have two more questions.
> > 1) The optimisation problem is formulated with a quadratic regulariser. Is it instead possible to use a linear regulariser? If so, do solutions obtained with a linear regulariser differ from those obtained with a quadratic regulariser?
> >
> > 2) The model uses normalisation of weights after each update. I suppose this is necessary for convergence and I do not have a problem with it. Nevertheless, my question is what are the consequences of not normalising the weights? Also, can good performance be achieved if weights are normalised only once every n>1 updates instead of every update?

---

> > > ### Author Response · Authors · 2024-08-09
> > >
> > > Thank you for the score bump. We're glad that our proposed revisions have improved the clarity of our work and we will incorporate them in our revised paper.
> > >
> > > In response to your questions:
> > >
> > > 1. This would not change the learned response distribution as this is set by $p_\text{target}$; however it would change the learning algorithm. Specifically, replacing the $L^2$ (quadratic) regularizer $\lambda||T({\bf s})||^2$ in equation (1) with an $L^1$ (linear) regularizer of the form $\lambda|T({\bf s})|$ would encourage sparser neural responses. However, to compensate, the interneurons would adapt to offset the $L^1$ regulariser. Now if the goal is to encourage sparsity, this could built into $p_\text{target}$. For example, rather than choosing $p_\text{target}$ to be Gaussian, it could be chosen to have heavier tails. In this case adding an $L^1$ regularizer to the objective would help the circuit achieve this goal.
> > > 2. The main reason for normalization is to prevent the weights from diverging (or collapsing), which is a notorious problem when the synapses update according to a Hebbian update without a homeostatic compensation mechanism; see, e.g., [Abbott & Nelson "Synaptic plasticity: Taming the beast", Nature Neuro. 2000]. As you suggest, this can be achieved by normalizing every $N>1$ steps. Alternatively, this step can be replaced by a dynamic normalization process similar to the one shown in appendix E.1 of [Duong et al. NeurIPS 2023].

---

> > > > ### Comment · Reviewer_Voaj · 2024-08-09
> > > >
> > > > Authors have replied thoroughly to my questions. I have appreciated the discussion around this paper.

---

> ### Author Response · Authors · 2024-08-12
>
> Likewise, we've appreciated your questions and engagement during the discussion period.
>
> Request: can you please edit your official score so that your score increase is reflected in the reported average? Thank you!

---

> > ### Comment · Reviewer_Voaj · 2024-08-13
> >
> > I updated the official score.

---

> > > ### Author Response · Authors · 2024-08-13
> > >
> > > Thank you!

---

### Official Review · Reviewer_zp8S · 2024-07-12

**Soundness:** 3
**Presentation:** 3
**Contribution:** 1
**Rating:** 5
**Confidence:** 3

**Summary:**

This paper introduces a method that utilizes multiple inhibitory neurons and the Hebbian learning rule to transform a signal into a representation that conforms to a specified target distribution. Specifically, the model incorporates Hebbian synaptic plasticity to establish connections that optimally match this target distribution. Interneurons within the model are adapted in terms of their gain and activation directions to respond most effectively to the input signals, mirroring the potential optimization mechanisms found in biological neural processing.

Utilizing this approach, the study processes natural images from the Kodak dataset through a wavelet transform. This extracts 2-dimensional signals, specifically pairs of wavelet coefficients for images at fixed horizontal spatial offsets ranging from d = 2 to d = 64. The model then transforms the distribution of this two-dimensional information into a Gaussian distribution. This transformation demonstrates the model's ability to handle complex data structures and align them with statistically predictable patterns, enhancing both the analysis and interpretation of natural images.

**Strengths:**

1. Clear Presentation: The paper articulates its concepts with exceptional clarity, facilitating a deep understanding of complex models and their applications.

2. Effective Transformation of Distributions: The model is highly effective at converting input signals into specific, targeted distributions, optimizing data for further processing and analysis.

3. Online Operational Capability: One of the model's significant advantages is its ability to operate online. This feature significantly enhances its practicality for real-world applications, allowing for real-time data processing and continuous learning without the need for retraining.

**Weaknesses:**

1.  A main concern is that the model is not biologically realistic. In addition to the authors' admission that the feedforward synaptic weights $W^T$ and feedback weights $-W$ are symmetrically constrained, there is also a question about the biological plausibility of using pairs of wavelet coefficients as inputs. It's unclear how biological systems would naturally derive such information and whether the distribution of these inputs is something that biological systems need to transform.

2. The setup of online learning with a batch size of 10 also raises questions about its biological feasibility. It's uncertain how biological systems could implement a similar mechanism.

3. The paper only demonstrates the method's effectiveness on two-dimensional inputs, which might not sufficiently prove its efficacy. In real-world scenarios, we often deal with inputs that are high-dimensional ($N>>2$), and the model's performance in such conditions remains untested.

4. The paper incorporates many specific settings, such as the choice of activation functions based on the Gaussianization of scalar signals. These particular choices may limit the generalizability and applicability of the model to different datasets or broader applications.

**Questions:**

1. As mentioned in the Weaknesses section, the model is not biologically realistic. Given this, my primary question is: What is the purpose of using neural networks to implement this function? Furthermore, if it does not realistically mimic biological processes, should it be compared with more engineering-oriented approaches?

2. Scaling of Inhibitory Neurons with Input Dimensionality: The model currently demonstrates with two-dimensional inputs, requiring three inhibitory neurons. How does the number of required inhibitory neurons scale as the dimensionality of the input increases? This is crucial for understanding the feasibility and complexity of the model when applied to higher-dimensional data.

**Limitations:**

Yes, the authors have addressed some of the limitations of their work in the Discussion section of the paper. They specifically mention the model’s lack of biological realism and its primary demonstrations within low-dimensional settings. These acknowledgments align with the checklist guidelines on discussing limitations.

---

> ### Author Rebuttal · Authors · 2024-08-07
>
> Thank you for your careful reading of our paper and for your thoughtful comments. We are pleased that you find the paper clear and appreciate the models online capabilities! We understand your concerns about the biological realism of our model and have addressed these concerns in our general author rebuttal above. Although there are aspects of our model that are not matched to underlying biological details, we do believe it offers a novel perspective for thinking about circuit computation that can provide a framework for interpreting experimental measurements.
>
> In response to weakness #4: We actually think this points to an interesting aspect of sensory signals. In some respects, it's quite surprising that the model can effectively Gaussianize two-dimensional signals just by reshaping the response marginals along a few axes. This is likely due to structure of natural signals, and the efficient coding hypothesis posits that sensory circuits take advantage of this as well. Therefore, while we agree there are aspects of our circuit that may not generalize to arbitrary datasets, we do not think this reduces its efficacy as a model of circuit computation.

---

> > ### Comment · Reviewer_zp8S · 2024-08-09
> >
> > I have carefully read the comments from other reviewers and appreciate the authors' detailed responses. I have a minor question regarding your discussion on the "Symmetry of synaptic weights" where you mention, "When the weights are decoupled, they converge asymptotically toward a symmetric solution due to the symmetry of their learning rules." While this conclusion is derived from both analytical and numerical analyses, I wonder if such symmetric connections are also observed in real biological systems, especially considering that this section addresses the question of biological plausibility.

---

> ### Author Response · Authors · 2024-08-09
>
> The most compelling example is the olfactory bulb, an early stage of olfactory processing in vertebrates, where excitatory mitral cells form dendrodendritic connections with local inhibitory granule cells [Shepherd 2003], leading to symmetric *connectivity* matrices (though not necessarily symmetric *weight* matrices). Within cortical circuits, it's unclear, though somatostatin interneurons that form local, reciprocal connections with pyramidal cells are promising candidates. We view the symmetry constraint as a testable prediction, which can potentially be measured, for example, in recently reported connectomics datasets from mouse visual cortex [Schneider-Mizell bioRxiv 2024].

---

> > ### Comment · Reviewer_zp8S · 2024-08-11
> >
> > Thank you to the authors for your thoughtful reply, which has deepened my understanding of this work. I will raise my score to a 5.

---

### Author Rebuttal · Authors · 2024-08-06

Thank you for your careful reading and helpful comments. Here we respond to two important concerns and provide individual responses below.

## Biological realism

Reviewers **zp8S**, **Voaj** and **txmi** listed the biological realism of our model as a primary concern that limits the applicability of our model in understanding neural circuits. We appreciate your concerns, though we still believe that our model represents a useful advance.

First, some of the concerns can be addressed with minor adjustments that do not affect the overall circuit computation. We list these concerns with the most easily addressed concerns first.
1. **Wavelet coefficients.** Apologies for the jargon.  These are responses of a local oriented filters (similar to Gabor filters) applied to natural images. They are qualitatively similar to simple cell responses in primary visual cortex [Field 1987], so they are representative of natural inputs to cortical circuits in the visual cortex. We will edit the text and refer to the inputs as "local filter responses''.
2. **Batch training.** Using a batch size 10 was an optimization choice and is not required. We could have implemented a fully online optimization algorithm with batch size 1. To demonstrate this, we've run Algorithm 1 in the fully online setting and reproduced Figure 3B of the main text; see the attached PDF.
3. **Symmetry of synaptic weights.** While Algorithm 1 enforces symmetry between the primary neuron-to-interneuron weights $W^\top$ and interneuron-to-primary neuron weights $-W$, this is not required. When the weights are decoupled, they converge asymptotically toward a symmetric solution due to the symmetry of their learning rules. This has been demonstrated both analytically and numerically in previous works; see, e.g, appendix E.2 of [Duong et al., NeurIPS 2023].
4. **Scalability.** We have strong reason to believe that the number of inhibitory interneurons required will scale reasonably with the dimension of the input. Visual inputs are highly structured; e.g., statistical dependencies between inputs rapidly decay with the distance between the inputs. Therefore, local interneurons only need to connect to neurons with overlapping or adjacent receptive fields, which greatly reduces the number of interneurons that are required as the size of the model increases.
5. **Violation of Dale's law.** This is the most serious concern (although note that Dale's law may not be as hard of a constraint as previously believed; see [Saunders et al. 2016, Granger et al. 2020]). There are two possible solutions. One approach is to seek a circuit implementation of the algorithm that obeys Dale's law. This could potentially be achieved by introducing additional (excitatory or inhibitory) neural populations, though the details of this construction need to be worked out. An alternative approach is to replace the current optimization problem with a *constrained* optimization problem that respects Dale's law and can be optimized via a projected gradient descent algorithm. As a preliminary test of this, we optimized the constrained optimization problem (in the context of Gaussianization of visual inputs) and find that the solution still significantly reduces the mutual information between neural responses (more than linear whitening, but less than the unconstrained model that violates Dale's law); see the attached PDF.

Second, we believe that our normative model strikes an effective balance between simplicity and biological realism that provides a useful framework for interpreting experimental measurements of neural circuits. In particular:
1. Our model conceptualizes the circuit computation as transforming the input distribution to achieve a target distribution (or set of distributions). This can potentially be tested experimentally by measuring distributions of circuit responses (and how these change when the input distribution shifts). We think this is especially useful when recording from large populations of neurons and there is some experimental work along these lines [Benucci et al. 2013].
2. We establish a link between the circuit level computation and physiological processes such as neural adaptation in local interneurons. This can be experimentally tested by measuring how interneuron FI curves adapt in response to changes in the distribution of circuit inputs.

## Relation to existing work

Reviewer **Voaj** pointed out that our original submission does not cite salient works related to circuit models that implement nonlinear transformations to efficiently encode information. We take this point quite seriously and will work to improve our description of the relationship of our work to related literature.  The following will be added to the paper:
> There are a number of existing computational models that explain how neural circuits can implement nonlinear transformations to efficiently encode their inputs. For example, there are several neural circuit models that implement forms of divisive normalization [Rubin et al. 2015, Chalk et al. 2017, Malo et al. 2024], a transformation that is optimal for efficient encoding of natural signals [Schwartz & Simoncelli 2001, Lyu 2011]. In addition, there is a body of work on normative spiking models derived from objectives which maximize the information encoded per spike [Koren & Deneve 2017, Alemi et al. 2018, Gutierrez & Deneve 2019], which can account for neural adaptation mechanisms such as gain control. Our work differs from these, by proposing a novel framing of sensory circuit computation in terms of transformations of probability distributions, which can be viewed as a population level version of the seminal work by Laughlin. We then demonstrate in a normative circuit model how interneurons can play a critical role in optimizing this objective by measuring the marginal distribution of circuit responses and adjusting their feedback accordingly.

---

### Author Response · Authors · 2024-08-12
**Thank you for the discussion**

Thank you for your thoughtful questions and comments, and for your engagement during the discussion period. Based on the discussion, we plan to revise our manuscript to expand our discussion on our model's **biological realism** and its relation to **existing models**. If any additional questions or comments come to mind, please let us know.

---

### Decision · Program_Chairs · 2024-09-25

**Decision:**

Accept (poster)

**Comment:**

The present study starts from the optimal transport objective function, and derives recurrent circuit dynamics with nonlinear interneurons to optimize the objective function. It links the normative computation and canonical cortical microcircuit, and technically it is a nonlinear extension of a recent circuit model for linear data whitening (Duong, et al., ICML 23; NeurIPS 23). Although the manuscript distinguishes the present study with Duong's papers, it would be better to discuss how much we gain by considering nonlinear interneurons from the perspective of computational performance and neuroscience. Lastly, please revise the manuscript accordingly based on the reviewers' feedback, including the discussions of Dale's law, symmetry of connection weight, scalability, etc.